

# A single-center retrospective study of ectopic lymphoid tissues in idiopathic membranous nephropathy: clinical pathological characteristics and prognostic value

Jing Zhang*, Siyu Chen*, Haiying Zheng, Siyi Rao, Yuanyuan Lin, Jianxin Wan and Yi Chen

Department of Nephrology, Blood Purification Research Center, The First Affiliated Hospital, Fujian Medical University, Fuzhou, Fujian, China
Fujian Clinical Research Center for Metabolic Chronic Kidney Disease, The First Affiliated Hospital, Fujian Medical University, Fuzhou, Fujian, China
Department of Nephrology, National Regional Medical Center, Binhai Campus of the First Affiliated Hospital, Fujian Medical University, Fuzhou, Fujian, China
* These authors contributed equally to this work.

Corresponding authors
Jianxin Wan, wanjx@fjmu.edu.cn
Yi Chen, chenyi@fjmu.edu.cn

## ABSTRACT

**Background:** In recent years, ectopic lymphoid tissue (ELT) has been increasingly confirmed as a new biomarker for kidney injury or inflammation. However, there is insufficient research on the relationship between ELT grading and the progression of idiopathic membranous nephropathy (IMN).

**Methods:** A total of 147 patients with biopsy-proven IMN in our institution from March 2020 to June 2022 were classified into five grades based on the different distribution of lymphocyte subsets in renal tissue (G0: no B cells or T cells, G1: scattered B and T cells, G2: clustered B and T cells, G3: an aggregation region of B and T cells without a central network, G4: highly organized and formed zones of B and T cells with a central network of follicular dendritic cells and scattered macrophages), and were further divided into low-grade group (G0+G1), intermediate-grade group (G2) and high-grade group (G3+G4). The clinicopathological data, induction treatment response and prognosis among the three groups were analyzed and compared retrospectively.

**Results:** As the grading of ectopic lymphoid tissues increased, patients were older, with a higher prevalence of hypertension, a higher 24-h urinary protein level, lower baseline hemoglobin and estimated glomerular filtration rate (eGFR) levels, and more severe renal pathological damage. Logistic regression analysis showed that after 6 months of induction treatment, patients in the high-grade group were more likely to be in non-remission than those in the low-grade group (odds ratios [ORs] of the three adjusted models were 4.310, 4.239, and 5.088, respectively, $P$-values were 0.005, 0.006, and 0.001, respectively). Kaplan-Meier survival analysis indicated that patients in the intermediate- and high-grade groups had significantly lower renal cumulative survival rate than those in the low-grade group ($P = 0.025$). Univariate Cox analysis showed that the risk of adverse renal outcome was 3.662 times higher in the intermediate- and high-grade groups than in the low-grade group (95% confidence

interval [CI] [1.078–12.435]; $P$ = 0.037). Multivariate Cox analysis revealed that failure of remission at the first 6 months (hazard ratio [HR] = 5.769; 95% CI [1.854–17.950]; $P$ = 0.002) remained an independent risk factor for poor renal outcome in patients with IMN.

**Conclusions:** Grading of renal ectopic lymphoid tissues correlates with disease activity and severity in IMN patients and can be used as an indicator to assess the risk of IMN progression.

# INTRODUCTION

Membranous nephropathy (MN) is one of the main causes of nephrotic syndrome (NS) in adults (*Ronco et al., 2021*). About 75% of cases are idiopathic MN (IMN), and the remaining 25% are secondary to certain known diseases, such as hepatitis B virus infection, systemic lupus erythematosus, malignant tumors, and medications (*Sinico et al., 2016*). Recent studies have shown that the prevalence of MN demonstrates an increasing trend in every age group (*Huang et al., 2022*; *Tang et al., 2017*). Patients with IMN often exhibit different clinical outcomes, with approximately one-third of patients experiencing spontaneous remission, one-third developing into non-progressive chronic kidney disease (CKD), and the remaining entering end-stage renal disease (ESRD) within 10 years (*Couser, 2017*). It has been reported that the renal survival rates of MN patients at 5, 10, and 15 years are 86%, 74%, and 56%, respectively (*Hoseini et al., 2023*). Therefore, identifying the risk factors affecting the outcomes of IMN patients and providing them with timely and effective treatment are crucial for improving the remission rate and prognosis.

Previous research has shown that diastolic blood pressure, age, gender, serum creatinine, uric acid, 24-h urinary protein, estimated glomerular filtration rate (eGFR), glomerulosclerosis, crescent formation, interstitial fibrosis, tubular atrophy, and vascular damage are risk factors for the progression from IMN to ESRD (*Lu et al., 2020*). In recent years, it has been found that tertiary lymphoid tissues (TLTs) can be used as new markers of renal injury and inflammation, as well as new therapeutic targets for renal diseases (*Luo et al., 2021*). Tertiary lymphoid tissues (TLTs), known as ectopic lymphoid tissues (ELTs) or tertiary lymphoide organs (TLOs), are organized immune cell clusters formed in non-lymphoid organs under chronic inflammatory conditions, which initiate adaptive immune response and coordinate local tissue immunity, and can also affect the severity, prognosis, and treatment response of various diseases including infections, autoimmune diseases, and cancer (*Sato, Tamura & Yanagita, 2023*).

ELTs are characterized by the formation of highly organized T/B cell aggregates with a network of follicular dendritic cells that support ectopic germinal center response (*Corsiero et al., 2016*). ELTs have been detected in various chronic kidney diseases, including IgA nephropathy (*Luo et al., 2021*), lupus nephritis (*Wang et al., 2023*), membranous nephropathy (*Wang et al., 2021*), and antineutrophil cytoplasmic antibody

(ANCA)-associated vasculitis (*Brix et al., 2018*), and are usually associated with progression of renal injury as well as poor prognosis. An early study by *Steinmetz et al. (2008)* proposed a new classification of lymphocyte clusters in lupus nephritis and ANCA-associated nephritis based on the different distributions of lymphocyte subsets in renal tissue. Similarly, we found that there were different distributions of B lymphocytes and other lymphocyte subsets in the renal tissue of IMN patients, manifesting as scattered B cells to highly differentiated B cell clusters and central follicular dendritic cell networks. More research is required to confirm the association between this different infiltration pattern of lymphocytes and clinicalpathological manifestations and prognosis of IMN. In this retrospective study, we included 147 patients with biopsy-proven IMN and divided them into three groups according to the infiltration of lymphocyte subsets in renal tissue. The correlations between the grading of ELTs and clinicalpathological indexes, 6-month treatment response, and prognosis in IMN patients were explored.

## MATERIALS AND METHODS

### Study patients

This cohort study included IMN patients who underwent renal biopsy from March 2020 to June 2022 at the Department of Nephrology, the First Affiliated Hospital of Fujian Medical University (Fuzhou, China). This study was approved by the Ethics Committee of First Affiliated Hospital of Fujian Medical University (MTCA, ECFAH of FMU [2015] 084-1) and conducted in accordance with the declaration of Helsinki. Informed written consent was obtained from all patients. Exclusion criteria: patients had less than six glomeruli in biopsy specimens; combined with other renal diseases (*e.g.*, diabetic nephropathy); received glucocorticoids or other immunosuppressants before renal biopsy; the presence of severe infections and ESRD at the time of renal biopsy; the follow-up duration less than 6 months. Finally, a total of 147 patients were included in this study (Fig. 1).

### Baseline data collection

Clinical parameters: gender; age; history of hypertension and type 2 diabetes mellitus; the levels of hemoglobin, serum albumin, serum creatinine, eGFR, total cholesterol, triglyceride, 24-h urinary protein, immunoglobulin G (IgG), and complement 3 (C3); the presence of microscopic hematuria; the positivity of serum anti-phospholipase A2 receptor (PLA2R) antibody.

Pathological indicators: (1) the proportions and extent of glomerulosclerosis, tubular atrophy, interstitial fibrosis, interstitial inflammatory cell infiltration, arteriolar lesion, and crescent formation under light microscopic; (2) the positivity of PLA2R, IgG, C3, and C1q under immunofluorescence; (3) chronicity index (CI) were evaluated according to the scoring method proposed by *Sethi et al. (2017)*.

### Grading of ectopic lymphoid tissues

We used CD20 staining as a marker for B lymphocytes, CD3 staining for T lymphocytes, CD4 staining for helper T cells, CD21 staining for follicular dendritic cells (fDCs) and CD68 staining for macrophages. Immunohistochemical staining was used to detect the

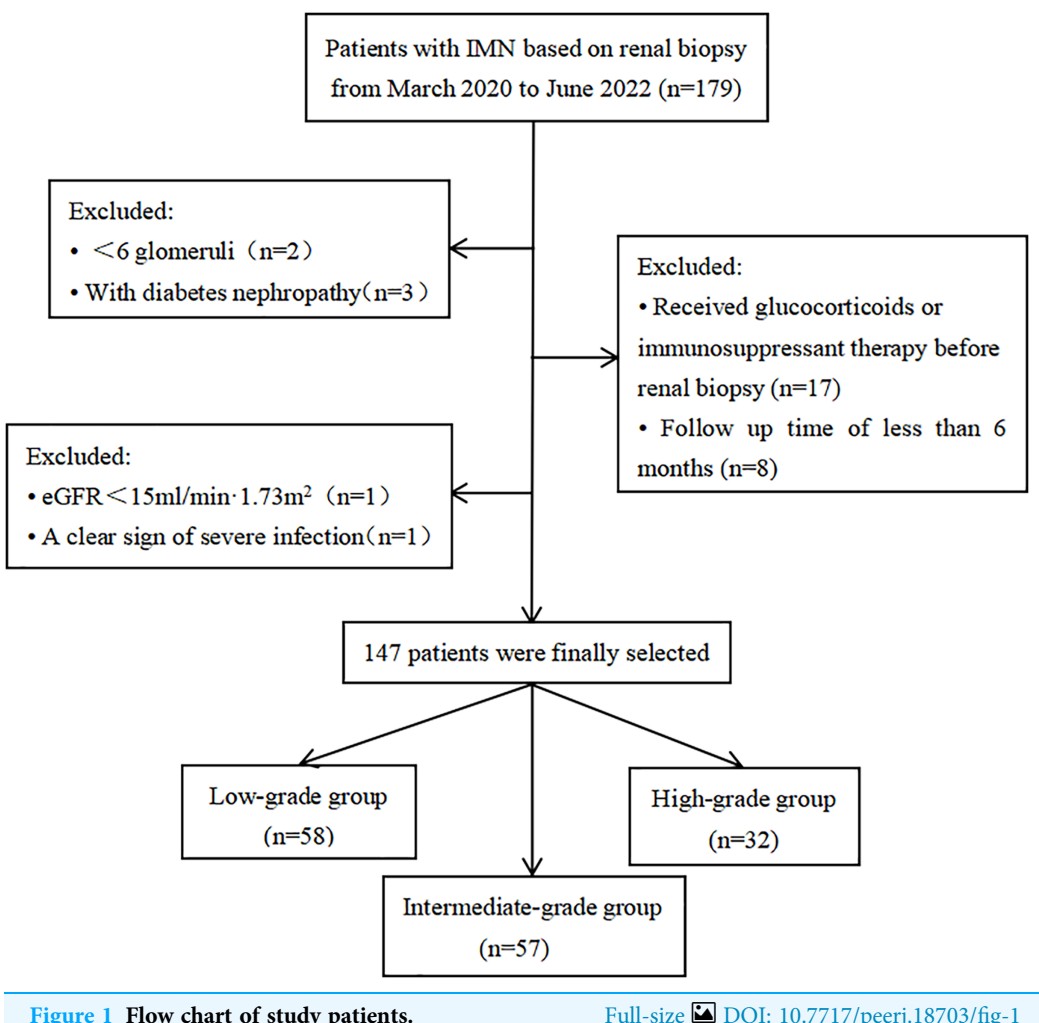

**Figure 1  Flow chart of study patients.**               

expression of CD3, CD4, CD8, CD20, CD21, and CD68 in renal tissue. Ectopic lymphoid tissues were classified into 0–4 grades according to the distributions of lymphocyte subsets. Grade 0: no CD20 positive B cells, CD3 positive T cells, CD4 positive Th cells, CD21 positive fDC, and CD68 positive macrophages in the renal tissue. Grade 1: CD20 positive B cells and CD3 positive T cells (including CD4 positive Th cells) were scattered in renal tissue, without CD21 positive fDCs and CD68 positive macrophages. Grade 2: CD3 positive T cells (including CD4 positive Th cells) and CD20 positive B cells were distributed in clusters without microanatomy separation, without CD21 positive fDCs and CD68 positive macrophages. Grade 3: an aggregation region composed of CD3 positive T cells (including CD4 positive Th cells) and CD20 positive B cells without a central network, without CD21 positive fDCs and CD68 positive macrophages. Grade 4: CD3 positive T cells (including CD4 positive Th cells) and CD20 positive B cells were highly organized and formed zones, with a central network of CD21 positive fDCs and scattered CD68 positive macrophages (Fig. 2).
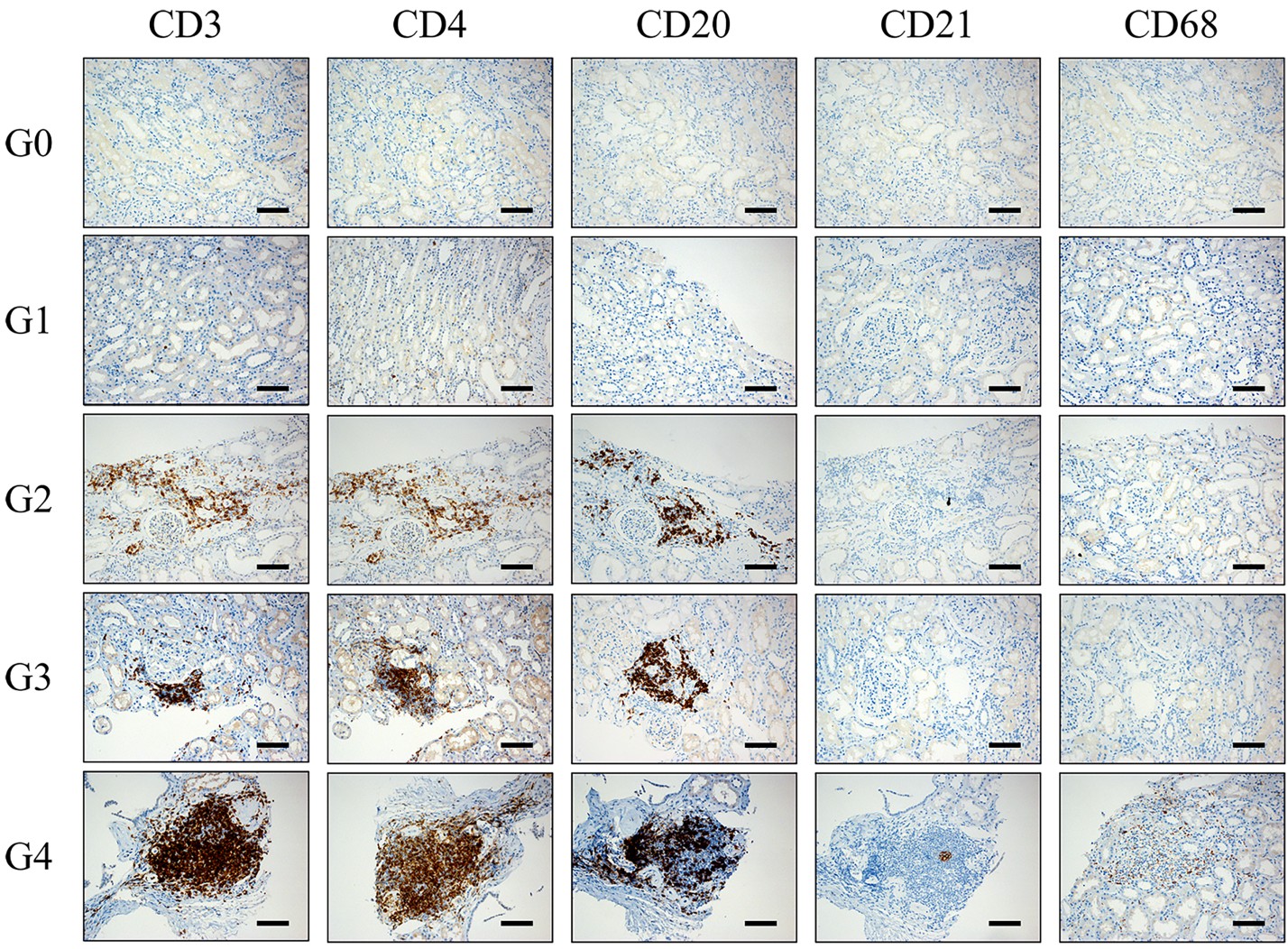

**Figure 2 Pathological manifestations of lymphocyte subsets distribution in renal tissue of IMN patients.** Grade 0: no CD20 positive B cells, CD3 positive T cells, CD4 positive Th cells, CD21 positive fDC and CD68 positive macrophages in the renal tissue. Grade 1: CD20 positive B cells and CD3 positive T cells (including CD4 positive Th cells) were scattered in renal tissue, without CD21 positive fDCs and CD68 positive macrophages. Grade 2: CD3 positive T cells (including CD4 positive Th cells) and CD20 positive B cells were distributed in clusters without microanatomy separation, without CD21 positive fDCs and CD68 positive macrophages. Grade 3: an aggregation region composed of CD3 positive T cells (including CD4 positive Th cells) and CD20 positive B cells without a central network, without CD21 positive fDCs and CD68 positive macrophages. Grade 4: CD3 positive T cells (including CD4 positive Th cells) and CD20 positive B cells were highly organized and form zones, with a central network of CD21 positive fDCs and scattered CD68 positive macrophages. fDCs: follicular dendritic cells. Th cells: T helper cells. Scale bar = 100 μm.

Due to the small sample size of our study, ELTs was categorized into three groups: low-grade group (G0+G1), intermediate-grade group (G2), and high-grade group (G3+G4).

## Induction treatments and follow-up

The main treatment regimen of IMN patients within 6 months after renal biopsy was recorded, including glucocorticoids, calcineurin inhibitors, cyclophosphamide, and other immunosuppressants, renin-angiotensin-aldosterone system (RAAS) inhibitors,

anticoagulants, lipid-regulating drugs, diuretics. The use of rituximab was also recorded during the entire follow-up period. All 147 participants were followed up until December, 2022, or reached the composite renal endpoint. Each patient was followed up for at least 6 months, with the levels of serum creatinine, serum albumin, and proteinuria being determined. Treatment response of the first 6 months was classified as complete remission, partial remission, or non-remission based on the aforementioned indicators.

## Definitions

Hypertension was defined as a systolic blood pressure ≥140 mmHg and/or diastolic blood pressure ≥90 mmHg on three measurements taken on different days at rest, or a history of hypertension, or using antihypertensive drugs to control blood pressure.

The eGFR was calculated using the Chronic Kidney Disease Epidemiological Collaboration (CKD-EPI) equation (*Levey et al., 2009*).

Complete remission (CR) was defined as proteinuria <0.3 g/24 h, along with normal range of serum albumin and serum creatinine. Partial remission (PR) was defined as proteinuria reduced to 0.3–3.5 g/24 h and >50% reduction from the time of inclusion, along with improved or normalized serum albumin and stable serum creatinine. Non-remission was defined as not meeting the criteria for complete and partial remission.

Composite renal endpoint was defined as persistent doubling of baseline serum creatinine or development of ESRD (eGFR < 15 ml/min/1.73 m$^2$ and initiation of dialysis), or death.

## Statistical methods

The data were performed using SPSS 25.0 software. All continuous variables underwent normal distribution tests, presenting as means ± standard deviations or as medians and interquartile ranges. The difference between three groups of normally distributed variables was determined by One-way ANOVA test, and the least significant difference (LSD) test was used for pairwise comparisons. The difference between three groups of non-normally distributed variables was determined by Kruskal-Wallis test. Categorical variables were presented as counts and percentages, and Chi-square test was used for comparisons among the three groups. Three logistic regression models were used to explore the relationship between the ELT grading and 6-month non-remission after induction treatment. Kaplan-Meier survival curves and log-rank test were used to analyze the association between ELT grading and the prognosis of IMN patients. Cox regression analysis was applied to explore the factors at basline that influence poor renal outcomes of IMN patients. Two-tailed $P < 0.05$ was considered a statistically significant difference.

## RESULTS

### Baseline characteristics of the study cohort

A total of 147 patients with IMN were included in this study, with an average age of 52.5 ± 13.6 years old, and 60.5% were male. According to the grading of ectopic lymphoid tissues, there were eight cases (5.4%) in G0, 50 cases (34.0%) in G1, 57 cases (38.8%) in G2, 21

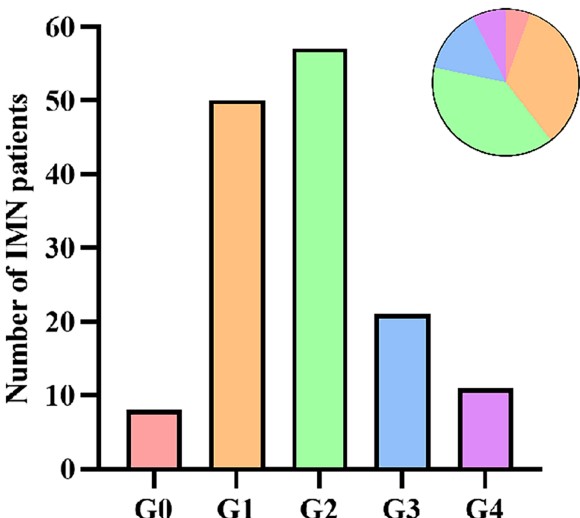

**Figure 3  Number of IMN patients according to ELT grading.**

cases (14.3%) in G3, and 11 cases (7.5%) in G4 (Fig. 3). Because of the small sample size of G0 and G4, we finally classified IMN patients into three groups: 58 cases (39.5%) in the low-grade group (G0+G1), 57 cases (38.8%) in the intermediate-grade group (G2), and 32 cases (21.7%) in the high-grade group (G3+G4). Compared with patients in the low-grade group, patients in the intermediate- and high-grade groups were older ($P = 0.004$), had a higher incidence of hypertension ($P = 0.006$), and with a lower baseline eGFR level ($P < 0.001$). Patients in the intermediate-grade group had a significantly higher 24-h urinary protein level than those in the low-grade group ($P = 0.015$). Patients in the high-grade group had a significantly lower baseline hemoglobin level than those in the low-grade group ($P = 0.005$). There were no statistically significant differences among the three groups in gender, the presence of microscopic hematuria, type 2 diabetes mellitus, and nephrotic syndrome, the positivity of serum anti-PLA2R antibody, and the levels of serum albumin, total cholesterol, triglyceride, creatinine, uric acid, serum IgG, and C3 (Table 1).

### ELT grading and pathological features of IMN patients

Compared with those in the low-grade group, patients in the intermediate-grade group had a higher proportion of glomerulosclerosis ($P = 0.002$). Patients in the high-grade group had a higher proportion of arteriolar lesions and more severe interstitial fibrosis and interstitial inflammatory cell infiltration than those in the low-grade group ($P = 0.004$, 0.011, 0.016, respectively). Patients in the intermediate- and high-grade groups had more severe tubular atrophy and higher chronicity index (CI) scores than those in the low-grade group ($P = 0.002$, <0.001). No significant differences were detected among the three groups in the proportion of crescent formation and the deposition of PLA2R, IgG, C3, and C1q in renal tissue (Table 2).

**Table 1 Clinical characteristics of IMN patients according to ELT grading.**

| | Low-grade group (n = 58) | Intermediate-grade group (n = 57) | High-grade group (n = 32) | P-value |
|---|---|---|---|---|
| Male, n (%) | 36 (62.1) | 38 (66.7) | 15 (46.9) | 0.178 |
| Age, years | 48.36 ± 13.07 | 53.67 ± 13.79[a] | 57.94 ± 11.98[a] | 0.004* |
| Hypertension, n (%) | 17 (29.3) | 32 (56.1)[a] | 18 (56.3)[a] | 0.006* |
| Type 2 diabetes mellitus, n (%) | 4 (6.9) | 8 (14.0) | 5 (15.6) | 0.351 |
| Nephrotic syndrome, n (%) | 23 (39.7) | 31 (54.4) | 18 (56.3) | 0.186 |
| Hemoglobin, g/L | 136.00 (124.00, 147.30) | 133.00 (117.50, 146.00) | 117.50 (105.00, 138.50)[a] | 0.005* |
| Serum albumin, g/L | 24.80 (22.03, 31.05) | 25.90 (21.70, 30.20) | 24.30 (21.00, 29.55) | 0.804 |
| Total cholesterol, mmol/L | 7.35 (5.68, 9.33) | 7.06 (5.36, 8.65) | 6.85 (4.52, 8.04) | 0.403 |
| Triglyceride, mmol/L | 1.81 (1.29, 2.76) | 2.15 (1.41, 3.92) | 1.98 (1.36, 2.97) | 0.375 |
| Uric acid, µmol/L | 385.80 (345.75, 428.65) | 363.80 (316.25, 472.65) | 375.85 (280.88, 463.88) | 0.848 |
| Serum creatinine, µmol/L | 64.65 (52.75, 80.25) | 67.30 (57.25, 87.40) | 72.30 (63.30, 101.75) | 0.063 |
| eGFR, ml/min/1.73 m$^2$ | 106.37 (91.70, 116.84) | 97.06 (83.06, 108.31)[a] | 81.15 (69.02, 99.15)[a] | <0.001* |
| 24-h urinary protein, g | 3.18 (1.24, 5.96) | 5.80 (2.59, 7.75)[a] | 5.53 (1.77, 8.00) | 0.015* |
| Microscopic hematuria, n (%) | 44 (77.2) | 41 (70.7) | 24 (75.0) | 0.887 |
| Serum PLA2R positivity, n (%) | 37 (63.8) | 40 (69.0) | 22 (68.8) | 0.233 |
| Serum IgG, g/L | 6.55 (4.04, 8.47) | 5.62 (3.91, 8.34) | 5.90 (4.61, 9.48) | 0.867 |
| Serum C3, g/L | 0.90 (0.77, 1.02) | 0.91 (0.78, 0.99) | 0.85 (0.74, 0.98) | 0.551 |

Notes:
Values for continuous variables are given as means ± standard deviation or median (interquartile range); values for categorical variables are given as count (percentage). eGFR, estimated glomerular filtration rate; PLA2R, phospholipase A2 receptor; ELT, ectopic lymphoid tissue.
* Two-tailed P < 0.05.
[a] P < 0.05 vs. low-grade group.

**Table 2 Pathological characteristics of IMN patients according to ELT grading.**

| | Low-grade group (n = 58) | Intermediate-grade group (n = 57) | High-grade group (n = 32) | P-value |
|---|---|---|---|---|
| Glomerulosclerosis, n (%) | 20 (34.5) | 38 (66.7)[a] | 19 (59.4) | 0.002* |
| Tubular atrophy | | | | |
| <25%, n (%) | 58 (100.0) | 51 (89.4)[a] | 27 (84.4)[a] | 0.002* |
| ≥25%, n (%) | 0 (0.0) | 6 (10.5)[a] | 5 (15.6)[a] | |
| Interstitial fibrosis | | | | |
| <25%, n (%) | 58 (100.0) | 53 (93.0) | 28 (87.5)[a] | 0.011* |
| ≥25%, n (%) | 0 (0.0) | 4 (7.0) | 4 (12.5)[a] | |
| Interstitial inflammatory cell infiltration | | | | |
| <25%, n (%) | 57 (98.3) | 52 (91.2) | 26 (81.3)[a] | 0.016* |
| ≥25%, n (%) | 1 (1.7) | 5 (8.8) | 6 (18.8)[a] | |
| Arteriolar lesions, n (%) | 6 (10.3) | 15 (26.3) | 13 (40.6)[a] | 0.004* |
| Crescent lesions, n (%) | 2 (3.5) | 7 (12.3) | 4 (12.5) | 0.177 |
| Chronicity index, score | 0 (0, 1) | 1 (0, 2)[a] | 2 (0, 3.8)[a] | <0.001* |
| PLA2R deposition, n (%) | 52 (89.7) | 53 (93.0) | 30 (93.8) | 0.735 |
| IgG deposition, n (%) | 56 (96.6) | 56 (98.3) | 32 (100.0) | 0.402 |
| C3 deposition, n (%) | 37 (63.8) | 37 (64.9) | 27 (84.4) | 0.096 |
| C1q deposition, n (%) | 11 (19.0) | 17 (29.8) | 13 (40.6) | 0.083 |

Notes:
Values for continuous variables are given as median (interquartile range); values for categorical variables are given as count (percentage); ELT, ectopic lymphoid tissue.
* Two-tailed P < 0.05.
[a] P < 0.05 vs. low-grade group.

**Table 3 Induction therapies and treatment response of IMN patients according to ELT grading.**

| | Low-grade group (n = 58) | Intermediate-grade group (n = 57) | High-grade group (n = 32) | P-value |
|---|---|---|---|---|
| Therapies during the first 6 months | | | | |
| No immunosuppressant, n (%) | 15 (25.9) | 8 (14.0) | 8 (25.0) | 0.248 |
| Glucocorticoids only, n (%) | 3 (5.2) | 3 (5.3) | 2 (6.3) | 0.975 |
| CNIs only, n (%) | 8 (13.8) | 8 (14.0) | 6 (17.8) | 0.794 |
| Glucocorticoids+CNIs, n (%) | 23 (40.4) | 21 (36.8) | 8 (25.0) | 0.363 |
| Glucocorticoids+CTX, n (%) | 1 (1.7) | 5 (8.8) | 4 (12.5) | 0.084 |
| Other immunosuppressants, n (%) | 8 (13.8) | 12 (27.1) | 4 (12.5) | 0.461 |
| RAAS inhibitors, n (%) | 51 (87.9) | 47 (82.5) | 28 (87.5) | 0.667 |
| Anticoagulants, n (%) | 49 (84.5) | 49 (86.0) | 26 (81.3) | 0.841 |
| lipid-regulating drugs, n (%) | 51 (87.9) | 50 (87.7) | 29 (90.6) | 0.908 |
| Diuretics, n (%) | 30 (51.7) | 26 (45.6) | 15 (46.9) | 0.793 |
| Treatment response | | | | |
| Non-remission, n (%) | 18 (31.0) | 23 (40.4) | 21 (65.6)[a] | 0.006* |
| Partial remission, n (%) | 24 (41.4) | 22 (38.6) | 5 (15.6)[a] | 0.036* |
| Complete remission, n (%) | 16 (21.6) | 12 (21.1) | 6 (18.8) | 0.568 |

Notes:
Values for categorical variables are given as count (percentage). ELT, ectopic lymphoid tissue; CNIs, calcineurin inhibitors; CTX, cyclophosphamide; RAAS, renin-angiotensin-aldosterone system.
* Two-tailed $P < 0.05$.
[a] $P < 0.05$ vs. low-grade group.

## ELT grading and induction treatment response

There were no statistically significant differences among the three groups for the treatment regimen applied during the first 6 months after renal biopsy. After 6 months of follow-up, 34 patients (23.1%) achieved complete remission, 51 patients (34.7%) achieved partial remission, and 62 patients (42.2%) demonstrated no remission. The partial remission rate was significantly higher in the low-grade group than that in the high-grade group ($P = 0.036$), the odds of non-remission was significantly higher in the high-grade group than that in the low-grade group ($P = 0.006$) (Table 3). For patients with nephrotic syndrome, the rates of immunosuppressant use in the low-, intermediate-, and high-grade groups were 91.3%, 83.9%, and 94.4%, respectively. The non-remission rate in the high-grade group was higher than those in the intermediate- and high-grade groups, but the differences were not statistically significant (Table S1).

We explored the relationship between the ELT grading and non-remission at 6-months using three logistic regression models. After correcting for the effect of sex, age, hypertension, eGFR, 24-h urinary protein quantification, and serum PLA2R positivity, the probability of non-remission in the high-grade group was 4.310 times higher than in the low-grade group (95% CI [1.537–12.089], $P = 0.005$). After correcting for the effect of glomerulosclerosis, tubular atrophy (<25% vs. ≥25%), interstitial fibrosis (<25% vs. ≥25%), interstitial inflammatory cell infiltration (<25% vs. ≥25%), arteriolar lesions, and crescent formation, the probability of non-remission in the high-grade group was 4.239 times higher than in the low-grade group (95% CI [1.527–11.768], $P = 0.006$). After correcting

**Table 4 Association analysis between non-remisson at 6 months and ELT grading.**

|  | Model 1 | | Model 2 | | Model 3 | |
| --- | --- | --- | --- | --- | --- | --- |
|  | OR (95%CI) | *P*-value | OR (95%CI) | *P*-value | OR (95%CI) | *P*-value |
| Low-grade group | Reference | | Reference | | Reference | |
| Intermediate-grade group | 1.551 [0.678–3.547] | 0.299 | 1.314 [0.577– 2.992] | 0.516 | 1.848 [0.816– 4.187] | 0.141 |
| High-grade group | 4.310 [1.537– 12.089] | 0.005* | 4.239 [1.527– 11.768] | 0.006* | 5.088 [1.934– 13.385] | 0.001* |
| *P* for trend | 0.020* | | 0.017* | | 0.004* | |

Notes:
ELT, ectopic lymphoid tissue; OR, odds ratio; CI, confidence; Model 1 adjusted for sex, age, hypertension, eGFR, 24-h urinary protein, and serum PLA2R positivity. Model 2 adjusted for glomerulosclerosis, tubular atrophy (<25% *vs.* ≥25%), interstitial fibrosis (<25% *vs.* ≥25%), interstitial inflammatory cell infiltration (<25% *vs.* ≥25%), arteriolar lesions, and crescent lesions. Model 3 adjusted for immunosuppressants, RAAS inhibitors, anticoagulants, lipid-regulating drugs, and diuretics.
* Two-tailed *P* < 0.05.

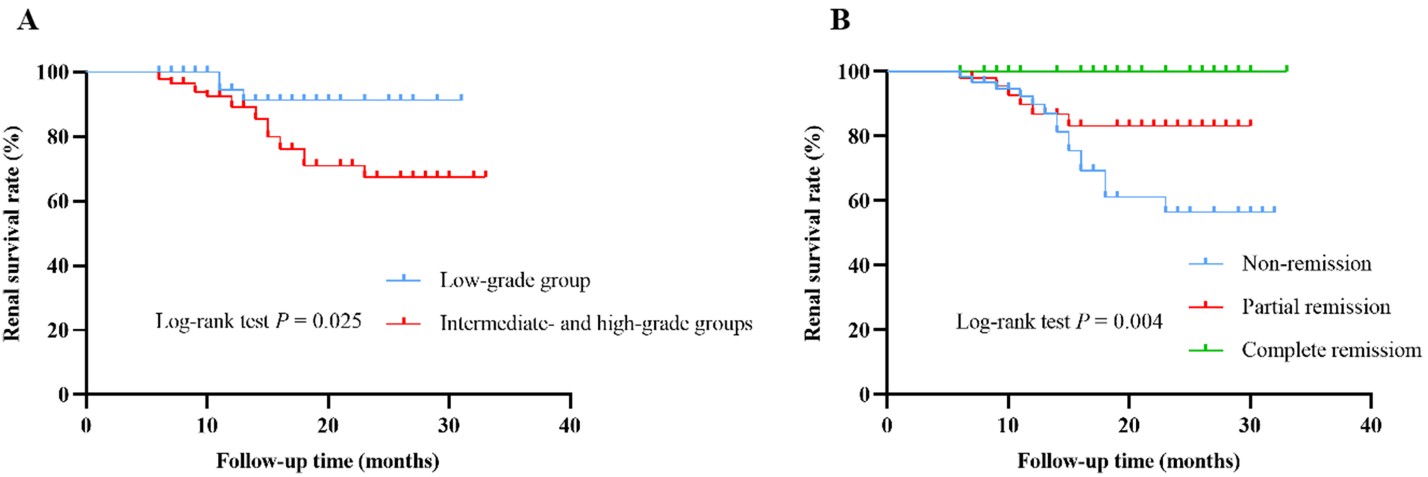

**Figure 4 Renal cumulative survival of IMN patients.** (A) Kaplan-Meier curves for renal cumulative survival according to ELT grading (low-grade group *vs.* intermediate- and high-grade groups). (B) Kaplan-Meier curves for renal cumulative survival according to remission status at 6 months (non-remission *vs.* partial remission *vs.* complete remission).

for the effect of immunosuppressants, RAAS inhibitors, anticoagulants, lipid-regulating drugs, and diuretics, the probability of non-remission in the high-grade group was 5.088 times higher than in the low-grade group (95% CI [1.934–13.385], *P* = 0.001) (Table 4).

### Renal survival analysis

During the median follow-up time of 14 (9, 21) months, 21 (14.3%) patients suffered from renal endpoint events, of which three patients entered ESRD (all in the high-grade group) and 18 experienced doubling of baseline serum creatinine (three in the low-grade group, 11 in the intermediate-grade group, and four in the high-grade group). Kaplan-Meier survival curves indicated that cumulative renal survival rate was significantly lower in the intermediate- and high-grade groups than that in the low-grade group (*P* = 0.025) (Fig. 4A). Patients in non-remission at 6-month had the lowest cumulative renal survival rate, followed by those in partial remission, and patients in complete remission had the highest cumulative renal survival rate (*P* = 0.004) (Fig. 4B). For patients with nephrotic syndrome, three entered ESRD (all in the high-grade group) and nine experienced

**Table 5 Cox regression analysis of factors that influence poor renal outcomes of IMN patients.**

| Variables | Univariate | | Multivariate | |
|---|---|---|---|---|
| | HR (95% CI) | *P*-value | HR (95% CI) | *P*-value |
| Age, years | 1.049 [1.010–1.090] | 0.014* | 1.011 [0.960–1.064] | 0.685 |
| Hypertension | 2.855 [1.150–7.088] | 0.024* | 1.457 [0.507–4.189] | 0.485 |
| Hemoglobin, g/L | 0.975 [0.954–0.998] | 0.031* | 0.999 [0.974–1.024] | 0.911 |
| eGFR, ml/min/1.73 m² | 0.968 [0.953–0.983] | <0.001* | 0.966 [0.939–0.995] | 0.021* |
| 24-h urinary protein, g/24 h | 1.111 [1.010–1.222] | 0.030* | 1.108 [0.977–1.256] | 0.111 |
| Serum PLA2R positivity | 1.239 [0.453–3.386] | 0.676 | | |
| ELT grading | 3.662 [1.078–12.435] | 0.037* | 1.200 [0.308–4.675] | 0.792 |
| Glomerulosclerosis | 2.248 [0.872–5.796] | 0.094 | | |
| Tubular atrophy | 3.281 [1.199–8.980] | 0.021* | 1.282 [0.357–4.609] | 0.704 |
| Interstitial fibrosis | 2.371 [0.696–8.077] | 0.168 | | |
| Non-remission at 6 months | 3.800 [1.472–9.810] | 0.006* | 5.769 [1.854–17.950] | 0.002* |
| Immunosuppressant therapy | 0.793 [0.265–2.372] | 0.679 | | |

Notes:
ELT grading: intermediate- and high-grade groups *vs*. low-grade group (reference); tubular atrophy: ≥25% lesions *vs*. <25% lesions (reference); interstitial fibrosis: ≥25% lesions *vs*. <25% lesions (reference).
* Two-tailed *P* < 0.05.

doubling of baseline serum creatinine (one in the low-grade group, six in the intermediate-grade group, and two in the high-grade group). The risk of entering ESRD was significantly higher in the high-grade group than those in the low- and intermediate-grade groups (*P* = 0.014) (Table S1).

We applied a Cox proportional hazards regression model to identify predictors affecting the poor prognosis of IMN patients. Univariate Cox regression analysis showed that age (HR = 1.049; 95% CI [1.010–1.090]; *P* = 0.014), hypertension (HR = 2.855; 95% CI [1.150–7.088]; *P* = 0.024), hemoglobin (HR = 0.975; 95% CI [0.954–0.998]; *P* = 0.031), eGFR (HR = 0.968; 95% CI [0.953–0.983]; *P* < 0.001), 24-h urinary protein level (HR = 1.111; 95% CI [1.010–1.222]; *P* = 0.030), ELT grading (HR = 3.662; 95% CI [1.078–12.435]; *P* = 0.037), tubular atrophy (HR = 3.281; 95% CI [1.199–8.980]; *P* = 0.021), and failure of remission at the first 6 months (HR = 3.800; 95% CI [1.472–9.810]; *P* = 0.006) were significantly associated with poor prognosis in patients with IMN. Multivariate Cox analysis revealed that failure of remission at the first 6 months (HR = 5.769; 95% CI [1.854–17.950]; *P* = 0.002) remained an independent risk factor for poor renal outcomes in patients with IMN, while high eGFR level (HR, 0.966; 95% CI [0.939–0.995]; *P* = 0.021) was an independent protective factor (Table 5).

## Patients receiving rituximab treatment and efficacy at 6 months

During the follow-up, a total of 22 patients (15.0%) received rituximab treatment, of which 30.3% (6/58) was in the low-grade group, 19.3% (11/57) was in the intermediate-grade group, and 15.6% (5/32) was in the high-grade group (Fig. 5A). The efficacy was assessed at 6 months following therapy, with total remission rate of 100% (6/6) in the low-grade group, 63.6% (7/11) in the intermediate-grade group, and 60% (3/5) in the high-grade

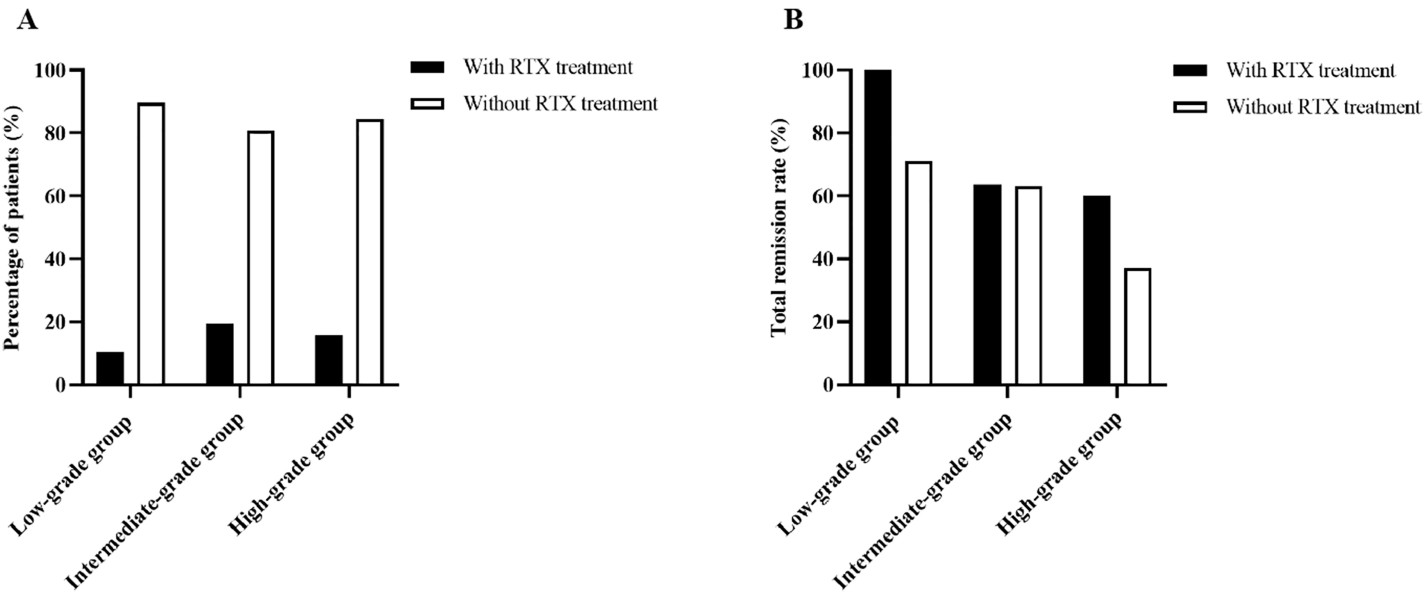

**Figure 5** **Percentage of patients receiving rituximab treatment and efficacy at 6 months.** (A) Percentage of patients receiving rituximab treatment according to ELT grading. (B) Total remisson rate with and without RTX treatment according to ELT grading.

group, which were higher than those of patients in the same group who did not use rituximab (the total remission rates were 71.2%, 63%, and 37% in the low-, intermediate-, and high-grade groups, respectively), but the differences were not statistically significant (Fig. 5B).

## DISCUSSION

In this retrospective cohort study, we graded the different distributions of lymphocyte subsets in renal tissue of patients with IMN. We found that patients in higher grade of ELTs were older, had a higher prevalence of hypertension, with a lower baseline hemoglobin level, more urinary protein excretion, worse renal function, more severe renal pathological damage, and poorer remission rate and prognosis compared to those in lower grade of ELTs.

ELTs usually develop at the site of inflammation and usually manifest from discrete clusters of small lymphocytes to complex structures resembling lymph nodes (*Pitzalis et al., 2014*). It is reported that ELTs promoted local autoimmunity through various pathways, including promoting local antigen presentation by B cells and other antigen-presenting cells, and providing critical survival signals for B cells and long-lived plasma cells, leading to B cell maturation and activation (*Pipi et al., 2018*). Our pathological observations revealed a diverse infiltration of B lymphocytes and other lymphocyte subsets in the renal tissue of IMN patients, ranging from scattered B cell clusters to highly differentiated B cell clusters and a central follicular dendritic cell network. We classified the ELTs into 0–4 grades (G0–G4) according to the distributions of lymphocyte subsets infiltration. We did not observe CD68-stained macrophage infiltration in renal tissue at G0–G3, while extensive macrophage infiltration in the renal interstitium

was observed at G4. As the grading of distribution of lymphocyte subsets in renal tissue increased, B and T lymphocytes infiltrated more deeply into the renal interstitium, presumably with more active cytokine secretion.

We found that the older the patient, the higher the ELT grade, suggesting an association between age and ELTs. *Ligon et al. (2020)* observed that B and T cells were highly enriched in the aged bladders and spontaneously formed organized bladder tertiary lymphoid tissues (bTLTs). An animal experiment also showed that older mice exhibited multiple TLTs after acute kidney injury, and suggested that TLT staging was a potential marker reflecting local injury and inflammation (*Sato et al., 2020*).

Anemia is a common complication of MN patients. It was reported that the prevalence of anemia was 23.8% when MN was diagnosed, and the cumulative prevalence was 50.7% during follow-up (*Li et al., 2023*). The main pathogenesis of renal anemia is erythropoietin (EPO) deficiency (*Nangaku & Eckardt, 2006*). Peritubular mesenchymal fibroblasts produce EPO in a hypoxia-induced manner. In chronic kidney disease (CKD), renal mesenchymal fibroblasts are converted to myofibroblasts, losing their ability to produce EPO but participating in renal fibrosis, leading to renal anemia (*Sato et al., 2019*). We found that as the grading of ELTs increased, the baseline hemoglobin level of IMN patients progressively decreased. It has been previously reported that chronic injury to renal tubules and interstitium leads to inadequate EPO synthesis and hence, anemia (*Babitt & Lin, 2012*). Thus, we hypothesized that the higher the grade of ELTs, the more severe the damage to the renal tubules and interstitium in patients with IMN.

It has been shown that renal ELTs may be related to anti-PLA2R antibody formation in IMN patients, and the proportion of anti-PLA2R autoantibodies was significantly higher in MN patients with renal ELTs than in those without ELTs (72.5% *vs.* 47.4%, $P < 0.001$) (*Wang et al., 2021*). Although our study did not find any association between renal ELT grading and the proportion of anti-PLA2R antibody formation, slightly higher positivity rates of serum anti-PLA2R antibodies and glomerular PLA2R antigen staining were observed in the intermediate- and high-grade groups compared to the low-grade group. Since our study was a single-center study with a small sample size, the relationship between serum anti-PLA2R antibodies and renal ELTs needs to be further investigated.

We found that patients with higher grade of ELTs had a higher prevalence of hypertension and a higher proportion of arteriolar lesions, with higher level of 24-h urinary protein and lower level of baseline eGFR compared to those with lower grade of ELTs. This suggested that the higher the ELT grade, the more likely that IMN patients are to have high blood pressure, increased urinary protein excretion and worsening renal function. Similarly, another study on MN also showed that patients with ELT formation had higher levels of serum creatinine, urinary protein and blood pressure (*Wang et al., 2021*). Moreover, we found that as the grading of ELTs increased, patients had more severe tubular atrophy, interstitial fibrosis, and interstitial inflammatory cell infiltration, and the proportion of glomerulosclerosis and arteriolar lesions tended to grow, the chronicity index scores tended to increase as well, suggesting that the ELT grading correlates with the degree of renal tissue damage. *Pei et al. (2014)* reported that the aggregation of renal interstitial inflammatory cells in IgAN patients was associated with decreased renal

function, excessive proteinuria, and severe glomerular, interstitial and arterial lesions, and patients with high-grade ELT exhibited more severe interstitial and arterial lesions, and a higher proportion of crescent formation. *Wang et al. (2023)* also found that in lupus nephritis, the presence of ELT was associated with severe tubulointerstitial inflammation, higher disease activity and chronicity index.

Several studies have reported that ELTs formation are associated with the progression of renal disease. *Lim et al. (2020)* revealed that ELTs formation could predict the development of adverse renal outcomes in crescentic glomerulonephritis. *Brix et al. (2018)* found that in ANCA associated vasculitis, patients with organized lymphocytic infiltrates in their biopsy had worse renal function during follow-up and were more likely to develop ESRD. Our study also demonstrated that renal ELT grading was associated with the poor prognosis of IMN patients, with cumulative kidney survival rate being significantly lower in patients in the intermediate- and high-grade groups than in those in the low-grade group, and all three patients who developed ESRD were in the high-grade group. Univariate Cox analysis revealed that the risk of adverse renal outcomes was 3.662 times higher in the intermediate- and high-grade groups than in the low-grade group (95% CI [1.078–12.435]; $P = 0.037$). However, Multivariate Cox analysis indicated that ELT grading was not an independent risk factor for renal survival in patients with IMN (HR, 1.200; 95% CI [0.308–4.675]; $P = 0.792$).

Treatment of IMN patients include supportive therapy and immunosuppressive therapy. Supportive therapy usually includes RAAS inhibitors, anticoagulants, lipid-regulating drugs, and diuretics. Immunosuppressive therapy include administering glucocorticoids, cyclophosphamide, calcineurin inhibitors, rituximab and other immunosuppressants. We found no statistically significant difference in the treatment regimen among the three groups of IMN patients during the first 6 months after renal biopsy. After 6 months of follow-up, the partial remission rate was significantly higher in the low-grade group than in the high-grade group ($P = 0.036$), and the non-remission rate was significantly higher in the high-grade group than in the low-grade group ($P = 0.006$), suggesting a correlation between the grading of renal ELTs and induction remission. Another study also confirmed that the rates of complete remission and total remission of proteinuria were significantly higher in patients without ELTs than in those with ELTs (*Wang et al., 2021*). Logistic regression was applied to further analyze the relationship between ELT grading and non-remission at 6 months. In the three calibrated regression models, it was observed that the probability of non-remission in patients in the high-grade group was significantly higher than that in the low-grade group (ORs were 4.310, 4.239 and 5.088, respectively, $P$-values were 0.005, 0.006, and 0.001, respectively), indicating that immunosuppressive therapy was less effective for patients in the high-grade group compared with those in the low-grade group. We therefore hypothesized that abnormal B cell infiltration and ELTs formation may involve in the pathogenesis of refractory IMN patients. Meanwhile, we found a correlation between remission status at 6 months and renal prognosis, with patients in complete remission having the highest cumulative renal survival, followed in descending order by patients in partial remission and those not in remission. Another study also evidenced that partial or complete remission of proteinuria

improved the prognosis of patients with IMN (*Zuo et al., 2013*), reiterating that early remission of IMN patients is crucial for improving the prognosis of IMN patients.

Rituximab (RTX) is a human-mouse chimeric monoclonal antibody that specifically binds to CD20 antigen on the surface of B cells, initiating an immune response to B cell lysis, preventing B cell proliferation and differentiation, reducing cytokine secretion, autoantibody and complement release to achieve a therapeutic effect (*Pescovitz, 2006*). The Kidney Disease: Improving Global Outcomes (KDIGO) clinical practice guidelines for glomerulonephritis, published in 2021, have recommended rituximab as a first-line treatment option for IMN (*KDIGO, 2021*). A growing number of studies have confirmed the safety and efficacy of rituximab, with approximately 60–70% of patients achieving partial or complete remission at 12 months on rituximab (*Gauckler et al., 2021*). In this study, 22 of 147 patients (15.0%) received rituximab treatment during follow-up, when the efficacy was assessed at 6 months, across the three groups, the total remission rate of patients using rituximab were higher than those of patients in the same group without rituximab treatment (100% *vs*. 71.2% in the low-grade group, 63.6% *vs*. 63% in the intermediate-grade group, 60% *vs*. 37% in the high-grade group, respectively), but the differences were not statistically significant. This suggests that early treatment with rituximab may improve the remission rate of IMN patients regardless of the grade of ELTs. Due to the small number of cases using rituximab in this study, the relationship between the distribution of lymphocyte subsets in renal tissue and the treatment response to rituximab in patients with IMN needs to be further investigated.

This study provides a new classification of the different distribution of lymphocyte subsets in renal tissue of IMN patients and explores its correlation with clinicopathological characteristics, treatment response and prognosis. However, this study has several limitations. Firstly, it was a single-center retrospective study with a relatively small sample size, therefore, we divided the patients with IMN into three groups (G0+G1, G2, and G3) instead of five groups (G0–G5). Secondly, this study only observed the remission of IMN patients at 6 months of treatment, but some patients may take longer than 6 months to achieve remission. Thirdly, due to the small number of cases using rituximab, the timing of receiving rituximab treatment (as a first- or second-line regimen) in patients with different grades of lymphocyte subpopulation distribution was not specifically described. Therefore, our results need to be further validated by prospective cohort studies with larger sample sizes and more extended follow-up.

## CONCLUSIONS

Renal ectopic lymphoid tissues grading correlates with disease activity and severity in IMN patients and can be used as a pathological indicator to assess the risk of IMN progression.

## ACKNOWLEDGEMENTS

The authors would like to thank the study patients and doctors at the Department of Nephrology, the First Affiliated Hospital, Fujian Medical University who contributed to this study.

### Funding

This study was supported by the Fujian Provincial Science and Technology Plan Project (No. 2021Y2005). The funders had no role in study design, data collection and analysis, decision to publish, or preparation of the manuscript.

### Grant Disclosures

The following grant information was disclosed by the authors:
Fujian Provincial Science and Technology Plan Project: 2021Y2005.

### Competing Interests

The authors declare that they have no competing interests.

### Author Contributions

- Jing Zhang conceived and designed the experiments, performed the experiments, prepared figures and/or tables, authored or reviewed drafts of the article, and approved the final draft.
- Siyu Chen conceived and designed the experiments, performed the experiments, prepared figures and/or tables, authored or reviewed drafts of the article, and approved the final draft.
- Haiying Zheng performed the experiments, prepared figures and/or tables, and approved the final draft.
- Siyi Rao performed the experiments, prepared figures and/or tables, and approved the final draft.
- Yuanyuan Lin performed the experiments, prepared figures and/or tables, and approved the final draft.
- Jianxin Wan analyzed the data, authored or reviewed drafts of the article, and approved the final draft.
- Yi Chen analyzed the data, authored or reviewed drafts of the article, and approved the final draft.

### Human Ethics

The following information was supplied relating to ethical approvals (*i.e.*, approving body and any reference numbers):

This study was approved by the Ethics Committee of First Affiliated Hospital of Fujian Medical University (MTCA, ECFAH of FMU [2015] 084-1).

### Data Availability

The raw data are available in the Supplemental File.

## Supplemental Information

Supplemental information for this article can be found online at http://dx.doi.org/10.7717/peerj.18703#supplemental-information.

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
