# Peer review of "A single-center retrospective study of ectopic lymphoid tissues in idiopathic membranous nephropathy: clinical pathological characteristics and prognostic value"

_PeerJ, doi:10.7717/peerj.18703_

## Round 0.1 · original submission · Major Revisions

More detailed analyses are needed to increase the value of the reported findings (please see reviewers' comments).

Reviewer 1 ·

Basic reporting

Satisfactory

Experimental design

Professional

Validity of the findings

Adds substantially to current knowledge on this field

Additional comments

I would like to congratulate the authors for their effort. Their research is interesting. The manuscript’s structure, table, and figure quality are satisfactory. Raw data were provided as expected. The methodology is presented in a clear, expert manner. Clearly, this study adds to current knowledge on ELT’s and their role in promoting renal injury in autoimmune glomerular diseases. I would advise the authors to focus on the following issues before resubmitting:
1. The English language needs improvement. Some examples can be found in rows 98-102, 139, 188-192, 196, 293, 363
2. In rows 134 and 323 there is probably a mistake. I suppose the authors mean calcineurin instead of calmodulin inhibitors. Prior to resubmitting, a clinician should do a thorough proofread of the manuscript.
3. Rituximab emerged as a first line agent for the treatment of MN long before KDIGO’s guideline update in 2021 (Fervenza et al. N Engl J Med. 2019 Jul 4;381(1):36-46. doi: 10.1056/NEJMoa1814427 , Seitz-Polski et al. Clin J Am Soc Nephrol. 2019 Aug 7;14(8):1173-1182. doi: 10.2215/CJN.11791018 , Dahan et al. J Am Soc Nephrol. 2017 Jan;28(1):348-358. doi: 10.1681/ASN.2016040449 , Fiorentino et al. Clin Kidney J. 2016 Dec;9(6):788-793. doi: 10.1093/ckj/sfw091). The authors should explain why they opted for treatments based on cyclophosphamide and CNI’s, as this could have affected the outcome, given the high positivity against PLA2R in the study population.
4. Given the fact that less than 60% of the patients were nephrotic, it would be helpful to have a subgroup analysis of their outcome, as they probably constitute the majority of utilized immunosuppressive therapy.
5. Examining the raw data, I noticed that hypoalbuminemia percentage is striking, although percentages of nephrotic syndrome are much lower. The authors should comment on that as well.

I wish you good luck with the publication of your manuscript!

Reviewer 2 ·

Basic reporting

No comment.

Experimental design

No comment.

Validity of the findings

Given the fact that the role and the pathophysiology of these structures remains uncertain, it would be interesting to compare the formation of ELT in cases of PLA2R+ membranous nephropathy with cases that are PLA2R- in order to see if there are differences in the clinical scenario, response to treatment and also prognosis.

Reviewer 3 ·

Basic reporting

This is a well written study with a detailed discussion written in a scientific manner. I especially like that the authors identified and discussed relevant study limitations.

I only have some small suggestions for improvement of the writing. For example, when the authors write: "We found that the age of patients with IMN tended to increase as the grade of ELTs increased", they likely mean that older patients have higher grade ELTs, but it is written as if the ELT grade influences the age of the patients.
I suggest the authors to more clearly state how the grading works in the abstract as i think most readers will not be familiar with the method.

Experimental design

The authors investigate a well-defined question. A downside is the single-center nature of the results. It seems for the Kaplan-Meier analysis (Fig, 4A), the grade-groups were grouped again into High (with intermediate)/Low? Please elaborate on why that was done and provide analyses with the original groups as well. As a follow up: Please indicate the numbers of events in the different groups.
It seems the high grade group can be defined only by the presence of CD68-positive cells. How is the outcome of patients when grouping them only based on CD68-status?
It seems that there might be some redundancy in the information provided through some of the stainings. Can the authors reduce the antibody panel (number of antibodies) in any other way to make applying their method more feasible?

Validity of the findings

In general, the authors took measures for validating their findings as far as possible within a retrospective single center study.
I do not see a data availability statement. While i think it is reasonable not to share the histology images, the raw table should be shared to enable replication of the results.

Additional comments

Please discuss that novel membranous nephropathy antigens (e.g. NELL1) have not been retrospectively investigated here.

---

## Round 0.2 · accepted · Accept

All the reviewers' comments were adequately resolved. No further comments.

Reviewer 1 ·

Basic reporting

No comment

Experimental design

No comment

Validity of the findings

No comment

Additional comments

Congratulations to the authors for their effort. Changes made in the manuscript are satisfactory.

Reviewer 2 ·

Basic reporting

no comment

Experimental design

The article has examined the development of ectopic lymphoid tissue in the setting of membranous nephropathy. While a causal link between the development of these structures and the diagnosis of membranous nephropathy cannot be formally established, it seems that older patients with more chronic lesions (arteriosclerosis, arteriolar hyalinosis, interstitial fibrosis and tubular atrophy) have a worse prognosis.

Validity of the findings

no comment

Reviewer 3 ·

Basic reporting

no comment

Experimental design

no comment

Validity of the findings

no comment

Additional comments

I thank the authors for addressing the issues i raised in my review.